# AutoMO-Mixer: An automated multi-objective multi-layer perspecton Mixer model for medical image based diagnosis

## Abstract

Medical image based diagnosis is one of the most challenging tasks which is criticala to human life. Accurately identifying the patient's status through medical images play an important role in treatment. Deep learning has achieved great success in medical image analysis. Particularly, Convolutional neural network(CNN) can obtain outstanding performance by learning the features in a supervised way. However, as there are too many parameters to train, CNN always requires a large scale dataset to feed, while it is very difficult to collect the required amount of images for a particular clinical problem. Recently, MLP-Mixer (Mixer) which is developed based multiple layer perceptron (MLP) was proposed, in which the number of training parameters is greatly decreased by removing convolutional layers in the architecture, while it can achieve the similar performance with CNN. Furthermore, obtaining the balanced outcome between sensitivity and specificity is of great importance in patient's status identification. As such, a new automated multi-objective Mixer (AutoMO-Mixer) model was developed in this study. In AutoMO-Mixer, sensitivity and specificity were considered as the objective functions simultaneously to train the model and a Pareto-optimal Mixer model set can be obtained in the training stage. Additionally, since there are several hyperparameters to train, the Bayesian optimization was introduced. To obtain a more reliable results, the final output was achieved by fusing the output probabilities of Pareto optimal models through the evidence reasoning (ER) approach. The experimental study on public datasets demonstrated that AutoMO-Mixer can obtain better performance compared with Mixer and CNN.

## 1 Introduction

With the development of modern medicine, medical image has become an essential way to carry out personalized and accurate diagnosis. Due to the outstanding image analysis ability, deep learning has been widely used in medical image based diagnosis and has achieved great success(Zhang & An, 2017)(Shen et al., 2017) in the past years.

As there are so many parameters to train in current deep learning model such as convolutional neural network (CNN)(Chua, 1998), a large scale dataset is always required to feed the model. In particular, the parameters in convolutional layer accounted a majority of the parameters. However, it is very difficult to collect sufficient medical image samples for a particular clinical problem in practice. Recently, a new deep learning model termed MLP-Mixer (Mixer)(Tolstikhin et al., 2021) which was developed based on multi-layer perception (MLP) was proposed. Compared with CNN, the convolutional layer is removed from Mixer, leading to decreasing the architecture parameter scale sharply and Mixer can be trained sufficiently through small scale dataset. On the other hand, Mixer can achieve similar performance with CNN(Tolstikhin et al., 2021). In summary, Mixer is a better architecture for building medical image based diagnostic model.

Furthermore, building the balanced model between sensitivity and specificity is necessary(Banerjee et al., 2018)(Mazurowski et al., 2008) in clinical diagnosis. Sensitivity indicates the proportion of people who are actually sick that are diagnosed as sick, whilst specificity indicates the proportion of people who are truly healthy who are diagnosed as healthy(Reitsma et al., 2005). As shown in

Table 1: An example of predicting distribution

|  | True-abnormal | True-normal |
|---|---|---|
| **Predicted-abnormal** | 35 | 3 |
| **Predicted-normal** | 7 | 55 |

Table 1, although the accuracy is high, sensitivity and specificity imbalance, resulting in higher rate of missed diagnosis. Therefore, a more balanced model is necessary and a multi-objective model which considers sensitivity and specificity simultaneously is needed.

As such, a new automated multi-objective Mixer (AutoMO-Mixer) model is developed in this study. In AutoMO-Mixer, both sensitivity and specificity are considered as the objective functions simultaneously and a Pareto-optimal model set can be obtained through the multi-objective optimization(Zhou et al., 2017) in training stage. In addition, since there are several hyperparameters which may affect the model performance in AutoMO-Mixer, Bayesian optimization(Pelikan, 2005) is used to train the hyperparameters. In testing stage, the Pareto-optimal models with balanced sensitivity and specificity are chosen so as to improve model diversity and stability. To obtain more reliable result, evidential reasoning (ER)(Yang & Xu, 2002) approach is used to fuse the output of selected Pareto-optimal models to obtain final outcome. The experimental studies on two public medical image datasets demonstrated that AutoMO-Mixer can outperform Mixer and CNN, and more balanced results can be achieved as well.

## 2 METHOD

### 2.1 FRAMEWORK

The framework of AutoMO-Mixer is shown in figure 1, which consists of training stage and testing stage. In training stage, medical images are fed into multiple Mixer models with randomly initialization. Then sensitivity and specificity are considered as the multi-objective functions simultaneously, and The Pareto-optimal Mixer model set can be obtained through multi-objective optimization. Meanwhile, since there are several hyperparameters which may affect the model performance, Bayesian optimization is used to optimize the hyperparameters. In testing stage, models with balanced sensitivity and specificity are fused using the ER approach to obtain more reliable final prediction model.

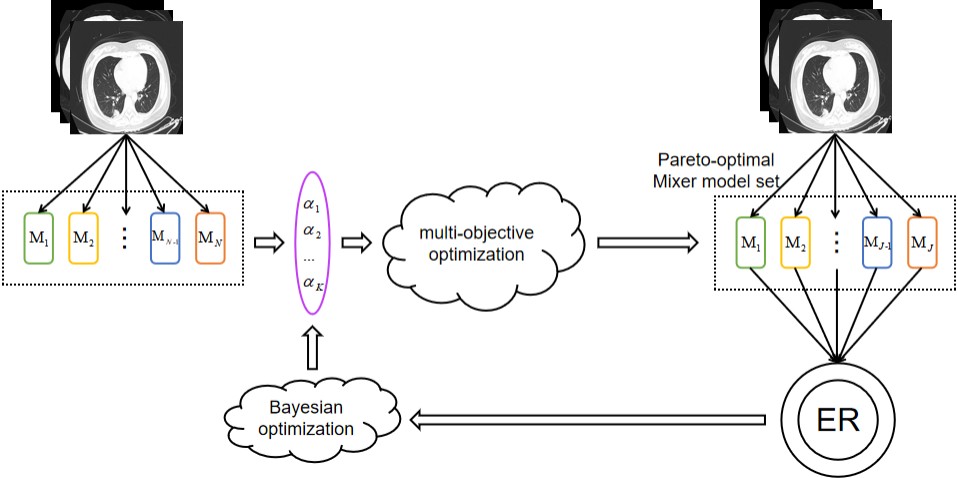

Figure 1: The framework of AutoMO-Mixer model.

## 2.2 TRAINING STAGE

Assume $\alpha = \{\alpha_1, ..., \alpha_K\}$ represents the hyperparameters, where K is the number of hyperparameter and $\beta$ denotes the parameters in Mixer models. $M = \{m_1, ..., m_q\}$ represents the Mixer model, where q is the number of models.

The MLP-Mixer network is a visual-oriented, all-MLP architecture as shown in figure 2. It consists consisting of four parts(Tolstikhin et al., 2021): splitting a picture into multiple patches shaped as a "patches×channels" table as an input; converting all patches into feature embedding with a set of fully connected networks; refining feature information through N mixer layers; and classifying them with a full-connected layer after global average pooling. Mixer is divided into token-mixer (feature refining along the columns direction of the table, i.e. spatial locations) and channel-mixer (feature refining along the rows direction of the table , i.e. channels), all using MLP for feature extraction, each MLP consists of two layers of full-connected and an activation function GELU.

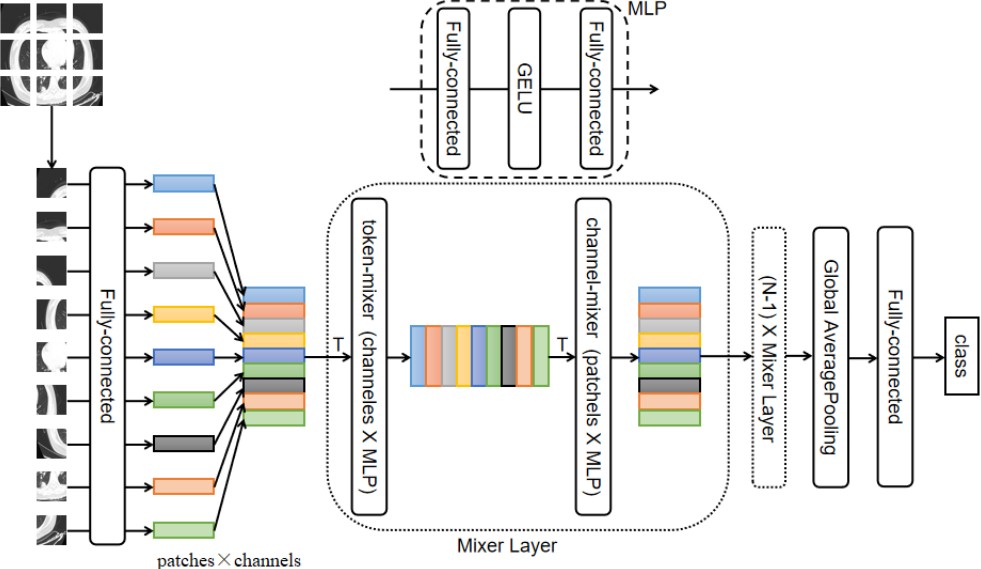

Figure 2: The structure of Mixer.

Since there are two groups' parameters to train, the training stage is a bilevel iterative optimization process. In other words, Mixers' model parameters and hyperparameters in AutoMO-Mixer are updated iteratively. The training stage is shown in figure 3. In the beginning, all the parameters are initialized randomly.

When the hyperparameters are determined, the Mixers' model parameters are trained first. To obtain the balanced model, sensitivity and specificity are considered as objective functions simultaneously, they are:

$$f_{sen} = \frac{TP}{TP + FN} \tag{1}$$

$$f_{spe} = \frac{TN}{TN + FP} \tag{2}$$

where TP is true positive, TN is true negative, FP is false positive, FN is false negative. The goal is to maximize $f_{sen}, f_{spe}$ simultaneously, that is:

$$f = \max_{\beta}(f_{sen}, f_{spe}) \tag{3}$$

To optimize the above function, an iterative multi-objective immune algorithm (IMIA)(Zhou et al., 2017)(Gong et al., 2008) is used. IMIA consists of six steps: initialization, cloning, mutation, deletion, update, and termination. First, the initial solution set denoted by $D(t) = \{M_1, ..., M_N\}$

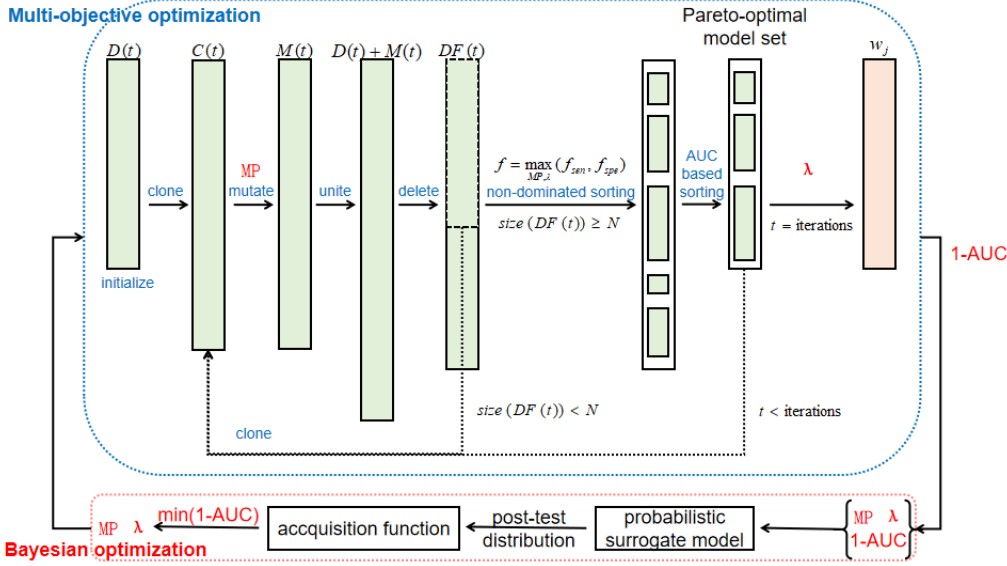

Figure 3: The illustration of training stage.

is generated, where $M_i = \{m_{i1}, ..., m_{iq}\}, i = 1, 2, ..., N$. Then the models with higher $M_i = \{m_{i1}, ..., m_{iq}\}, i = 1, 2, ..., N$ will be replicated using the proportional cloning method. In the third step, a probability of mutation is randomly generated for each model, and the model performs mutation when its probability is larger than the mutation probability. After the mutation, the new models are generated. If the models have same sensitivity and specificity, only one model is remained and the all the other models are removed. Then the model size is kept through AUC based non-dominated sorting strategy. This training process will not stop until the maximum number of iterations is reached.

Then the Pareto-optimal Mixer model set is generated, where the model set size is J. Now the optimal parameters for Mixer are obtained and hyperparameters are optimized. Since there are hyperparameters in Pareto-optimal model selection and fusion phase, the weight for each Pareto-optimal model should be estimated, which is denoted by $w_j$. As the balanced model between sensitivity and specificity is desired, the ratio between them is considered, which is $\frac{f_{sen}}{f_{spe}}$ or $\frac{f_{spe}}{f_{sen}}$. When the ratio is less than 0.5 or greater than 1, the model is considered as extreme imbalance, setting $w_j$ as 0. The expression of $w_j$ is as follows:

$$
w_j = \begin{cases} \lambda \frac{f_{sen}^j}{f_{spe}^j} + (1 - \lambda)AUC, & when\ 0.5 \leq \frac{f_{sen}^j}{f_{spe}^j} \leq 1 \\ \lambda \frac{f_{spe}^j}{f_{sen}^j} + (1 - \lambda)AUC, & when\ 0.5 \leq \frac{f_{spe}^j}{f_{sen}^j} \leq 1, j = 1, 2, ..., J, \\ 0 & Other\ situations \end{cases} \tag{4}
$$

where $\lambda$ indicates the importance of balance, and 1-$\lambda$ indicates the importance of AUC. After calculating the $w_j$ for each model, the weights are normalized, that is:

$$
\sum_{j=1}^{J} w_j = 1, 0 \leq w_j \leq 1 \tag{5}
$$

Then the Pareto-optimal models are fused through ER approach. Assume there are J models in Pareto-optimal model set, and the output probability (normal and abnormal) for each model is denoted by in $P_j = \{P_j^1, P_j^2\}, j = 1, 2, ..., J$, where $P_j^1$ is the output probability of abnormal, and $P_j^2$ is the output probability of normal. The final output probability $P_{fin}^c, c = 1, 2$ is obtained through the ER fusion strategy(Wang et al., 2006), that is:

$$
P_{fin}^c = ER(P_j^c, w_j), j = 1, 2, ..., J, c = 1, 2 \tag{6}
$$

where ER is:

$$P_{fin}^c = \frac{\mu \times [\prod_{j=1}^{J}(w_j P_j^c + 1 - w_j) - \prod_{j=1}^{J}(1 - w_j)]}{1 - \mu \times [\prod_{j=1}^{J}(1 - w_j)]}, c = 1,2 \tag{7}$$

The normalized factor $\mu$ is:

$$\mu = [\sum_{c=1}^{2}\prod_{j=1}^{J}(w_j P_j^c + 1 - w_j) - \prod_{j=1}^{J}(1 - w_j)]^{-1} \tag{8}$$

The hyperparameter $\alpha$ can be optimized through Bayesian optimization(Pelikan, 2005)(Springenberg et al., 2016), where the objective function is:

$$f_H = \min_{\alpha}(1 - AUC) \tag{9}$$

Eqs. (3) and (9) are optimized iteratively until it reaches the termination critation.

### 2.3 TESTING STAGE

An illustration of testing stage is shown in figure 4. The medical images are fed into the trained Pareto-optimal Mixer models and the output probabilities from each model can be obtained. Then ER approach is used to fuse the outputs of all the Mixer models and final output can be obtained.

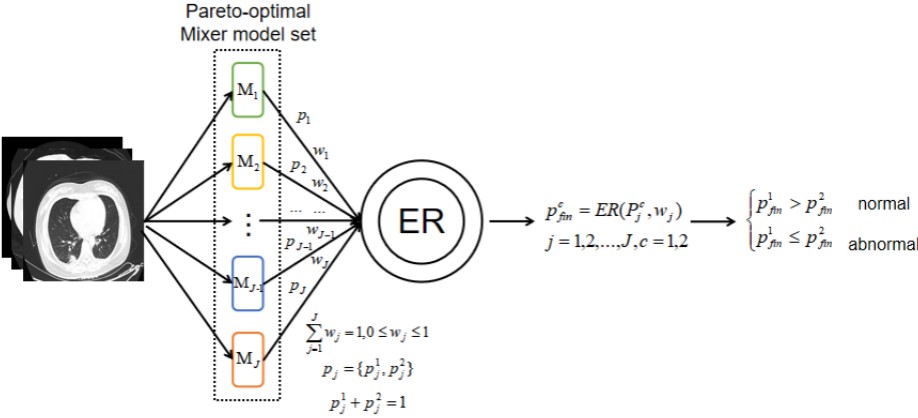

Figure 4: The illustration of testing stage.

## 3 EXPERIMENTS

### 3.1 MATERIALS AND SETUP

The two public databases used in this paper are MRI images of brain tumors(Panigrahi, 2021) and CT images of COVID-19(LuisBlanche, 2020). The first dataset aims to help people for detecting brain tumors. It contains MRI scans of the brain, and each type of brain has 1500 images. We selected 900 images for modeling. The next dataset is collected from COVID19-related papers from medRxiv, bioRxiv, NEJM, JAMA, Lancet, etc. It contains 349 CT scans that are positive for COVID-19, This dataset can be used to perform classification and automatically detect COVID-19 on CT scans. The distribution of the two types of data is shown in Table 2. The examples of two datasets are shown in figure 5.

Before feeding the model, all the images are resized into 224 x 224 images, and each image is divided into 14 x 14 patches, where each patch size is 16 x 16. There are two hyperparameters,

Table 2: Distribution of two types of data

|  | Train | Train | Test | Test |
|---|---|---|---|---|
|  | **Abnormal** | **Normal** | **Abnormal** | **normal** |
| **Brain Tumor** | 360 | 360 | 90 | 90 |
| **COVID-19** | 280 | 318 | 69 | 79 |

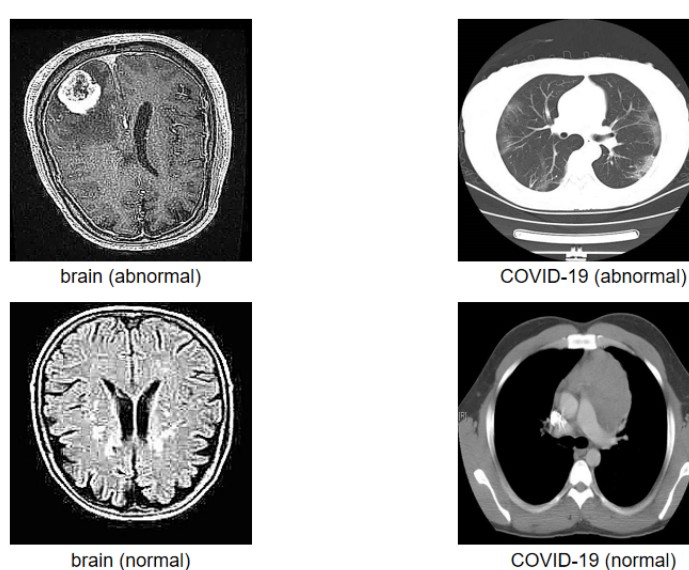

Figure 5: The examples of two datasets.

MP and $\lambda$, which the range is set between 0 and 1. The network structure contains five parameters, they are: the number of Mixer layers, the number of channels(Hidden size C), the number of hidden layer neurons of token-mixing(MLP dimension Ds), and the number of hidden layer neurons of channel-mixing(MLP dimension Dc). The settings are shown in table 3.

Table 3: The range of values for Mixer network structure parameters

| parameters | range of values |
|---|---|
| Number of layers | [2, 3, 4] |
| Hidden size C | 256*[1, 1.2, 1.4, 1.6] |
| MLP dimension Ds | 196*[2, 3, 4, 5] |
| MLP dimension Dc | 256*[2, 4, 6, 8, 10] |

In comparative study, Mixer and CNN model are evaluated. For fairness of comparison, the number of layers of the two models is required to be equal. We make the number of layers five. The four parameters of the Mixer network are set to 5, 256, 392, 1024, respectively.

AUC(Zhao et al., 2011) is an ideal indicator to evaluate the reliability of the prediction results, and the larger the AUC shows that the better the classification of the model. Accuracy (ACC) represents the correct rate of diagnosis and is also an important indicator for evaluating the model. Therefore, sensitivity (SEN), specificity (SPE), AUC, ACC are used for evaluation. All the experiments are performed five times and the average results are shown.

## 3.2 RESULTS

The evaluation results on two datasets are shown in table 4, 5 and figure 6, respectively. It can be seen that Mixer can achieve the similar performance, while the number of parameters to train is greatly less than CNN. Furthermore, AutoMO-Mixer outperforms both Mixer and CNN in all four evaluation results, which means that AutoMO-Mixer is a more promising model.

Table 4: The results on Brain Tumor dataset

|  | SEN | SPE | AUC | ACC |
|---|---|---|---|---|
| Mixer | 0.92 | 0.92 | 0.96 | 0.92 |
| CNN | 0.92 | 0.91 | 0.95 | 0.92 |
| **AutoMO-Mixer** | **0.93** | **0.93** | **0.97** | **0.93** |

Table 5: The results on COVID-19 dataset

|  | SEN | SPE | AUC | ACC |
|---|---|---|---|---|
| Mixer | 0.75 | 0.72 | 0.80 | 0.73 |
| CNN | 0.71 | 0.70 | 0.77 | 0.70 |
| **AutoMO-Mixer** | **0.83** | **0.80** | **0.88** | **0.81** |

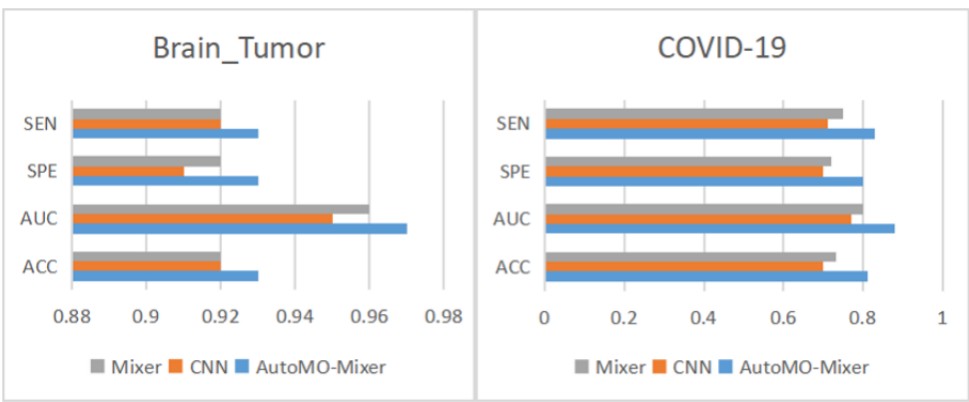

Figure 6: The performance of three models.

## 4 CONCLUSIONS

In this study, a new model termed as AutoMO-Mixer was developed for image based diagnosis. In AutoMO-Mixer, sensitivity and specificity are considered as the objective functions simultaneously and a Pareto-optimal Mixer model set can be obtained. Meanwhile, Bayesian optimization is used to train the hyperparameters to improve model performance. In addition, to obtain more reliable results, the Pareto-optimal models with a balance between sensitivity and specificity are selected and the ER approach is used to fuse their output for results. The experimental results on two public medical image datasets showed that AutoMO-Mixer can outperform Mixer and CNN. Furthermore, more datasets will be used to validate model performance.

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
