# OpenReview forum: "AutoMO-Mixer: An automated multi-objective multi-layer perspecton Mixer model for medical image based diagnosis"
_ICLR.cc/2022/Conference — ICLR 2022 Submitted_

### Official Review · Reviewer_Bur6 · 2021-10-28

**Correctness:** 1
**Technical Novelty And Significance:** 1
**Empirical Novelty And Significance:** 1
**Recommendation:** 1
**Confidence:** 5

**Main Review:**

In my view, this paper makes several technical and conceptual errors, and fails to make any noteworthy contributions. The paper is difficult to read and skips over many important details necessary to understand and reproduce the work. It also contains numerous typos and grammatical errors, including a misspelling of “perceptron” in the title.

MOTIVATION
In the introduction, an argument is made to justify the choice of MLP-Mixer for small-scale datasets, in which it is stated “the convolutional layer is removed from Mixer, leading to decreasing the architecture parameter scale sharply and Mixer can be trained sufficiently through small scale dataset.” I don’t believe this to be true - to my knowledge it has not been shown that MLP mixers are more efficient in terms of number of parameters than CNNs, in fact research on transformers and MLP-Mixers shown that the lack of inductive bias in these models make them less efficient than CNNs. They only benefit when model capacity is large and the dataset is very large. In the original MLP-Mixer paper it was necessary to pre-train on ImageNet21k and Google’s proprietary JFT-300M dataset with 300M images to make MLP-Mixer competitive. The whole rationale for selecting MLP-Mixer for this task seems incorrect.

RELATED WORK
There is no related work section. The paper fails to communicate how this work relates to the current state of the art in medical image classification, other works using MLP-Mixers, or Bayesian Optimization for hyperparameter tuning.

My next concern is about the goal of balancing sensitivity and specificity with Pareto-optimal models. It is not clear to me why a balance of sensitivity and specificity is indeed a good thing in general, and it has not been convincingly argued by the authors. I can think of counter-examples in medicine, e.g. in breast cancer screening different countries set different goals of recall rates because they value FP and FN differently. Why should we try to balance these if they do not have the same cost to society?

TECHNICAL DESCRIPTION & CONTRIBUTIONS
The description of the approach (AutoMO-Mixer) does not provide enough detail to understand or reproduce the work. The overall scheme depicted in Figure 1 is confusing. It is stated that this figure depicts a training and test stage, but this is not evident from the figure. The main content of the methodology is a recap of the MLP-Mixer architecture, along with a depiction of it in Figure 2. On page 3 it is stated that there are “2 groups” to train, but it is not clear what these two groups are. I am assuming that these are two groups of MLP-Mixers, one being the Pareto-optimal set? The “Multi-objective” goal which gives the model it’s name is really a single objective - namely to optimize for a balanced model defined in Eq. 3. This is achieved through an evolutionary algorithm which operates on a group of MLP-Mixer models, a strange choice. It would make more sense to build this objective into the training of the MLP-Mixer itself. The authors also claim some novelty by using a Bayesian optimization of hyperparameters but this is also problematic - first, they are using an off-the-shelf BayesOpt solution and secondly the hyperparameters they are optimizing for are just the hyperparameters of the Pareto-optimal model selection. It has nothing to do with training the MLP-Mixers.

EXPERIMENTS
The experimental results are also concerning. To start with, the experiments are conducted on extremely small datasets from Kaggle - a set of 3000 MRI images and a set of 700 Covid CT scans. The MLP-MIxers were trained with random initialization - when dealing with such small datasets the standard procedure should be to use transfer learning. The architecture used in not one of the standard specifications defined in (Tolstikhin et al, 2021). A comparison is made against stand-alone MLP-Mixers and CNNs, and to keep the comparison “fair” the number of layers is chosen to be the same. The correct setup would be to choose similar #parameters or FLOPS. The experiments report the sensitivity and specificity along with accuracy, but never do an ablation to show how performance is affected by class imbalance which is very relevant for sensitivity and specificity.

CLARITY & ERRORS
Finally, there are many instances where explanations are unclear or terms are misused. For example, Table 3 purports to list the MLP-Mixer parameters, but they have in fact listed the hyperparameters. The parameters of the model are of course the tunable weights. They hyperparameters of the multi-objective optimization which are optimized for are states as MP and \lambda, but it is never stated what MP is. Figure 3 and Figure 1 are very difficult to understand even after reading the paper.

**Summary Of The Paper:**

This work proposes AutoMO-Mixer, an approach for combining multiple MLP-Mixer models for medical image classification. In short, the paper builds on recent work in MLP-Mixer models, describing an ensemble learning scheme designed to balance sensitivity and specificity.

**Summary Of The Review:**

The motivation for this work appears to be flawed, the technical description is unclear and lacks novelty, there are flaws in the experimental design, and the paper contains several errors and lacks clarity.

---

### Official Review · Reviewer_rxpX · 2021-11-02

**Correctness:** 2
**Technical Novelty And Significance:** 2
**Empirical Novelty And Significance:** 2
**Recommendation:** 3
**Confidence:** 4

**Main Review:**

Strengths:
1. Obtaining sufficient annotated medical data for training data-hungry neural networks is currently a challenge. The paper proposed an alternative to CNNs, MLP-Mixer, which requires less training data.
2. Usually, medical datasets are not balanced, which leads to an imbalance in sensitivity and specificity. The authors propose to use a multi-objective immune algorithm to balance between sensitivity and specificity.
3. Results seem promising (but comparisons to some other related approaches might be missing).
4. Using Evidential Reasoning (ER) to fuse model set is interesting (but lacks comparison to simple voting-based fusion strategies).

Weaknesses:
1. “In summary, Mixer is a better architecture for building medical image-based diagnostic model.” - The authors make a very strong claim without providing any reference or evidence to back up the claim.

2. The technical novelty of the paper is limited. The proposed method is the fusion of MLP-Mixer, Bayesian optimization, and ER. However, application in the medical domain is important.

3. The notation is confusing, and the method section is hard to follow. ‘N’ denotes the number of layers of the second paragraph in 2.2 and is used later for D(t). Does N represent the number of layers of models or the number of models in the initial set D(t)?

4. M represents the Mixer model, which is a set consisting of ‘q’ models. It’s hard to follow what M represents.

 5. It’s is not evident that the better performance of AutoMO-Mixer is because of multi-objective function or better hyper-parameters with Bayesian optimization.

6. The use of Evidence Reasoning (ER) to fuse the probabilities of different models is not well justified.

7. Literature review and background information about EV and IMIA are missing.

8. Figure 2 is taken/inspired from the original MLP-Mixer paper. Authors should mention this in the caption with referencing.

Minor Points:
1. “Then the models with higher M_i.” It is not clear what does high means in this context.

2. I would suggest changing the caption of the figures to make it self-explanatory. Authors should add a gist of the figure or a few lines about the information being visualized or conveyed.

3. The authors may want to consider moving background information about MLP-Mixer in section 2.2 (Training stage) to a new section (literature review/background). It breaks the flow while understanding the training strategy of the proposed algorithm.

4. There are typos in the paper (and in the title!).


Suggestions for improvements:

1. Imbalance in sensitivity and specificity is mainly due to the imbalance dataset. Thus, it’s essential to compare the performance with training with classes weighted according to the number of samples in the training set and Focal loss [1] to justify using the multi-objective function and multi-objective immune algorithm.

2. Authors should define ‘Pareto-optimal’ as the audience might not be familiar with the terminology.

3. Authors should compare it with other fusing strategies (averaging, voting, etc.) to justify the use of Evidence Reasoning (ER) to fuse the probabilities of different models.

4. In Table 4, I would also suggest fine-tuning hyperparameters of the CNN and Mixer model with Bayesian optimization to empirically demonstrate that the better performance of AutoMO-Mixer is because of multi-objective function, and not because of better hyper-parameters with Bayesian optimization.

5. “In summary, Mixer is a better architecture for building medical image-based diagnostic model.”  Though Mixer-MLP has fewer parameters than CNN, they don’t need to be the best of medical images. I suggest authors run experiments on a large-scale medical dataset such as The Cancer Genome Atlas (TCGA) [2] to back up such a strong claim.

[1] Lin TY, Goyal P, Girshick R, He K, Dollár P. Focal loss for dense object detection. InProceedings of the IEEE international conference on computer vision 2017 (pp. 2980-2988).

[2] Hutter C, Zenklusen JC. The cancer genome atlas: creating lasting value beyond its data. Cell. 2018 Apr 5;173(2):283-5.


**Summary Of The Paper:**

The authors proposed a new method for learning imbalanced medical image datasets. They propose to use the recently propose MLP-Mixer instead of CNN  to reduce the number of trainable parameters and thus can be trained with fewer images as compared to CNNs.
To handle the class imbalance in the datasets, they use multi-objective loss functions to balance between sensitivity and specificity. Bayesian optimation is used for finding the optimal hyperparameters. The proposed method is evaluated on two medical image datasets, showing improvements over CNN and Mixer baselines.


**Summary Of The Review:**

While the work shows promise, particularly in the medical imaging domain, more experiments and comparisons with existing methods (as mentioned above) are required to improve the paper further and justify the use of EV and multi-objective function to handle the imbalance between sensitivity and specificity.

---

### Official Review · Reviewer_5jFR · 2021-11-02

**Correctness:** 3
**Technical Novelty And Significance:** 2
**Empirical Novelty And Significance:** 2
**Recommendation:** 3
**Confidence:** 4

**Main Review:**

The main strengths of the paper are in recognizing the importance of targeted solutions towards smaller data sets, through introducing and evaluating a convolutional layer-free neural network architecture. The attempted balancing of the final sensitivity and specificity metrics by weighing individual Mixer models using an iterative multi-objective immune (genetic) IMIA algorithm, is also relatively novel.
However, the paper also has a number of concerning issues:
1. The standard method for balancing sensitivity/specificity with deep learning models, is to the best of our knowledge, selecting the corresponding operating threshold/point on the model output on a held-out validation dataset. It might be clarified as to whether this was performed in evaluating the (single) Mixer and CNN models for the results as presented in Tables 4 and 5, especially since the CNN sensitivity/specificity values appear approximately balanced.
2. While the goal for the Mixer models is stated to be maximizing f_sen and f_spe simultaneously, the max(.) function used in Equation 3 appears to have the default definition of taking the maximum value amongst the two values, which does not appear the intended meaning. The equation might thus be clarified if appropriate, e.g. f = min(abs(f_sen – f_spe)) if balancing their values is the priority.
3. For IMIA, it is stated that models with higher M_i = {m_i1 … m_iq} will be replicated. The definition of m_iq might be clarified; are they model performance by AUC, and if so, was some training data further held out for internal validation of such metrics?
4. It is claimed that the Mixer model set is Pareto-optimal (Page 4). This Pareto-optimal property might be more explicitly proven.
5. The Mixer model set size is stated to be J (Page 4). The value and selection process for J might be described, for the experiments attempted.
6. MP is stated to be one of the two hyperparameters for IMIA together with lambda, however MP does not appear to be introduced or reference anywhere else in the paper. Is MP actually J?
7. The network structure is then stated to consist of five parameters. However, only four parameters are then described, and listed in Table 3.
8. The fairness of the comparison between (single) Mixer and CNN models was stated to be imposed by the number of layers being equal in both models (Page 6). However, the structure and training details of the CNN models are not indicated. It is therefore difficult to determine as to whether the CNN architecture used was reasonable, and whether the potential of the CNN models had been maximized.
9. The individual contributions (i.e. Mixer architecture, IMIA, ER fusion) were not disentangled through proper ablation experiments. As such, it is difficult to determine the main source of the performance improvement for AutoMO-Mixer, and indeed whether some of these contributions were necessary. For example, there appears no reason why CNN models could not be combined by IMIA, or even through basic ensembling, for comparison purposes.
10. The results are stated to be the average of five independent runs (Page 6). The standard deviation/confidence interval attained might thus be considered to be presented.
11. There are a number of spelling & grammatical issues, e.g. “perspecton” in the title, “developed based (on) multiple layer perceptron” in the abstract, “accounted (for) a majority” in the introduction section, etc.



**Summary Of The Paper:**

This paper proposes an AutoMO-Mixer framework for medical image diagnosis, consisting of several features: Firstly, the use of the MLP-Mixer deep learning model architecture, which is stated to achieve reduced parameter scale via the elimination of convolutional layers, thus making the architecture more suited to the training of small-scale datasets. Secondly, the learning of a (Pareto-optimal) Mixer model set, by weighing the multiple objectives of achieving a balanced sensitivity/specificity, and overall performance as measured by AUC. Thirdly, the fusion of the outputs of these selected models in the Mixer model set, by an ER fusion strategy. The AutoMO-Mixer framework was compared against a single Mixer alone and a CNN with equivalent number of layers, on two publicly-available CT images datasets on brain tumor and COVID-19 detection respectively. While all three approaches performed similarly on the brain tumor classification dataset, AutoMO-Mixer was shown to perform appreciably better than the (single) Mixer and CNN approaches, on the COVID-19 dataset.


**Summary Of The Review:**

While the motivations of the paper are well-founded, there remain multiple significant concerns, in particular over the methodological details and the comprehensiveness of the evaluation.

---

### Official Review · Reviewer_SKkc · 2021-11-02

**Correctness:** 2
**Technical Novelty And Significance:** 1
**Empirical Novelty And Significance:** 1
**Recommendation:** 1
**Confidence:** 4

**Main Review:**

# Strengths
I cannot find a significant strength from this study.

# Weakness:
1. No novelty. Everything looks a direct application of existing technique.
2. Poorly motivated. Why MLP Mixer should be tried for medical images? Why multi-objective optim should be introduced here? Even if the experiments show the "proposed" model looks better, so what? I cannot find any specific design for medical images.
3. Lack of empirical results. Only "CNN", "Mixer" and the so-called "AutoMO-Mixer" is compared, why? There are so many variants in every part...
4. Bad presentation. Writing and figures needs improvement. No story in this paper.

**Summary Of The Paper:**

This study is a simple combination of multiple-objective optimization and MLP Mixer on medical image application, with poor experiments to support any conclusion. Everything looks a direct application of existing technique. I don't understand why this study is submitted to a top conference.

**Summary Of The Review:**

A simple combination of existing techniques, with poor experiments to support any conclusion. No story.

---

### Decision · Program_Chairs · 2022-01-20

**Decision:**

Reject

**Comment:**

The reviewers recommended a rejection. The authors of the paper did not respond.